# Diastology and MitraClip^®^ Outcomes: Multicenter Real-World Evidence Study

**DOI:** 10.3390/medicina61061092

**Published:** 2025-06-16

**Authors:** Vivek Joseph Varughese, Chandler Richardson, James Pollock, Patryk Czyzewski, Hata Mujadzic, Michael Cryer

**Affiliations:** 1Internal Medicine, Prisma Health, School of Medicine, University of South Carolina, Columbia, SC 29201, USA; chandler.richardson@prismahealth.org (C.R.); james.pollock@prismahealth.org (J.P.); patryk.czyzewski@prismahealth.org (P.C.); 2Cardiology, Prisma Health, School of Medicine, University of South Carolina, Columbia, SC 29201, USA; hata.mujadzic@prismahealth.org (H.M.); michael.cryer3@prismahealth.org (M.C.)

**Keywords:** functional MR, MitraClip^®^, heart failure, diastology

## Abstract

*Background and Objectives*: MitraClip^®^ (MC) placement has been extensively used as an intervention for mitral transcatheter edge-to-edge repair (mTEER) for functional mitral regurgitation (FMR). The aim of our study is to analyze the association between the pre-procedural echocardiographic parameters of diastolic function (DF) and one-year outcomes after MC placement. *Materials and Methods*: The study was designed in a retrospective longitudinal cross-sectional format. In total, 224 patients who underwent MC placement between January of 2021 and March of 2024 were included in the study. The Primary Efficacy Endpoint (PEE) was determined by an absence of heart failure hospitalizations requiring Intravenous Diuretics or cardiac-related death in the one-year follow-up period. Multivariate regression analysis was carried out to identify the pre-procedural echocardiographic parameters of DF that had a significant association with a failure to reach the PEE. A two-tailed *p*-value < 0.05 was used to determine statistical significance. *Results*: Of the 224 patients included in the study, 85 patients (37.94%) failed to reach the PEE or had worsening symptoms. The mean mitral valve (MV) deceleration time was 176.88 ms (164.14–189.62) in the symptom-worsening group compared to 201.53 ms (186.01–217.07) in the symptom-improvement group. The mean of the E/A ratio (MV peak E velocity/A velocity) was noted to be 2.35 (1.97–2.74) in the symptom-worsening group compared to 1.90 (1.68–2.13) in the symptom-improvement group. After multivariate regression analysis, the E/A ratio was found to have a significant association with a failure to reach PEE: Odds Ratio (OR): 1.61 (1.13–2.29), *p*-value: 0.008. The area under the curve (AUC) analysis for the E/A ratio was calculated at 0.603 (0.50–0.69) for the symptom-worsening group. *Conclusions*: Patients that failed to reach the PEE had a lower pre-procedural MV deceleration time of 176.88 ms (164.14–189.62); however, no association was observed between MV deceleration time and a failure to reach the PEE in the multivariate regression analysis. The pre-procedural E/A ratio had a significant association with symptom worsening after multivariate regression analysis: OR: 1.61 (1.13–2.29). The AUC for the E/A ratio in the symptom-worsening group was 0.603, making it a more moderate predictor than random guessing for the failure to reach the PEE.

## 1. Introduction

Functional mitral regurgitation (FMR) stems from defects in the coaptation of the mitral leaflets due to cardiac remodeling, leading to an abnormal papillary muscle interaction with the leaflets, not necessarily due to anatomic defects of the mitral valve (MV) apparatus. This can be from wall motion abnormalities and papillary muscle dysfunction related to prior myocardial infarction; cardiac remodeling related to long-standing heart failure; and/or concomitant left atrial dilatation, most commonly from long-standing atrial fibrillation [1]. FMR can be reversible through remodeling achieved through guideline-directed medical therapy (GDMT), revascularization, and resynchronization strategies. Regurgitant Fraction > 50%, Regurgitant Volume > 60 mL, Vena Contracta ≥ 0.7 cm, and Effective Regurgitant Orifice Area (EROA) ≥ 0.4 cm^2^ are the echocardiographic parameters defining severity in FMR [2]. The 2020 AHA/ACC guidelines on the management of valvular heart diseases recommend having an expert in advanced heart failure management as an integral part of the treatment team for FMR. Mitral transcatheter edge-to-edge repair (mTEER) is a transcatheter approach to mechanically fix the malcoaptation of mitral leaflets caused due to annular dilatation in FMR. The current guidelines recommend it as a treatment modality in severe symptomatic FMR with persistent symptoms despite maximally tolerated GDMT, and an ejection fraction (EF) between 20% and 50%, assuming the pulmonary artery systolic pressure (PASP) remains less than 70 mm Hg and the left ventricular end systolic dimension (LVESD) remains less than 7 mm Hg [2]. Surgical mitral valve repair or replacement is recommended in patients who are not candidates for mTEER [2].

FMR is a consequence of cardiac remodeling due to advanced heart failure; at the same time, is also a factor worsening cardiac remodeling, creating a vicious cycle. This makes the management challenging, often creating a gray area between the switch from medical management to interventions. This is analogous to the situation of ischemic cardiomyopathy management, where trials like the STITCH and COURAGE compared medical management to revascularization strategies [3,4]. COAPT and the MITRA FR were two of the landmark clinical trials that studied outcomes for mTEER [5,6]. RESHAPE- HF2 was a later trial that proved superior outcomes in the mTEER plus GDMT group in terms of one year heart failure hospitalizations and death, as well as an improvement on the baseline Kansas City Cardiomyopathy Questionnaire (KCCQ) scores [7]. With advanced heart failure strategies, including assist devices, continuous inotropic infusions, and heart transplantations, becoming more robust and prevalent, the selection of patients experiencing a meaningful difference in heart failure symptoms through mTEER becomes relevant. Insights from the COAPT and MITRA FR trials gave rise to the disproportionate MR theory [8] and of the predictive value of the EROA to the LVEDV (Left Ventricular End Diastolic Volume) ratio in predicting mTEER outcomes. Advanced 3D imaging techniques involving the Vena Contracta area to quantify the regurgitant jet, especially in cases of eccentric jets associated with FMR, have been developed [9].

There have been retrospective studies in the past studying global longitudinal strain post procedure as an outcome predictor after mTEER, but there is a general paucity of data analyzing the impact of diastolic function in FMR [10]. Possible explanations include the overestimation of the MV peak E velocity due to higher trans-mitral pressure gradient created due to the regurgitant jet, a lower A velocity, and therefore an overestimation of the E/A ratio, which is important for analyzing diastolic function in reduced ejection fraction states [5]. However, irrespective of the regurgitation severity, pulmonary vascular congestion due to elevated end diastolic pressures is the primary symptom driver in FMR. In this context, the extent of diastolic dysfunction can have a potential impact on meaningful symptomatic improvement with mTEER, as LA and LV compliance can have major effects on end diastolic pressures. The aim of our study is to analyze the association between the pre-procedural echocardiographic parameters of the LV diastolic function and one-year outcomes after MitraClip^®^ placement.

## 2. Materials and Methods

### 2.1. Study Design

The study was designed in a retrospective longitudinal cross-sectional format. Patients who underwent mTEER using MitraClip^®^ were selected for the study. Patients who underwent the procedure at Prisma Health Richland Hospital, Prisma Health Baptist Hospital, and Prisma Health Greenville campus between January of 2021 and March of 2024 were selected for the study.

Inclusion Criteria: Patient aged > 18, who underwent MitraClip^®^ placement for SMR, also known as functional mitral regurgitation (FMR), with a 3+/4+ severity of mitral regurgitation.

Exclusion Criteria: Patient undergoing MitraClip^®^ placement for PMR. Patients without one-year surveillance echocardiogram and KCCQ score documentations were excluded from the study. Patients with a coexisting diagnosis of terminal illness like malignancies, and patient death in the one-year follow-up period due to non-cardiovascular-related conditions, were excluded from the study. Patients with advanced heart failure treatment options like LVAD in the one-year follow-up period were excluded from the study. Patients requiring repeat clip placements/requiring surgical valve repair in the one-year follow-up period were excluded. Patients with comorbid mitral stenosis were excluded from the study.

After a careful manual chart review of 326 patients, 224 patients were included in the study. One-year outcomes were analyzed for the selected patients according to the criteria defined by the Mitral Valve Research Consortium on efficacy and safety endpoints.

As per the recommendations of the mitral valve research consortium, Primary Efficacy Endpoint was defined by the absence of hospitalizations for heart failure/volume overload in the one year following mTEER placement requiring IV diuretics or cardiovascular-related deaths in the one-year post MitraClip^®^ placement. Safety endpoints were also defined as per the criteria specified by the mitral valve research consortium defined as ER/hospital visits in the 30 days following mTEER placement for diagnoses that could be attributed to the procedure.

Study subjects were divided into three groups:The symptom-improvement group (Group 1): patients who had a change in their baseline KCCQ (Kansas City Cardiomyopathy Questionnaire) score > 20 at one year, an absence of heart failure hospitalization/death, and a <50% increase in their preprocedural diuretic usage.The stable-symptom group (Group 2): patients with no heart failure hospitalization or death in the one-year follow-up period, but with a change in their baseline KCCQ score < 20 or increase in their diuretic usage > 50%.The symptom-worsening group (Group 3): patients who experienced heart failure-related hospitalizations or death in the one-year follow-up period.

### 2.2. Methodology and Statistics

Study subjects were divided into three groups as per the one-year outcomes specified in the study design. The first part of the statistical approach was determining the pre- and post-procedural echocardiographic dimensions indicative of the LV diastolic function, with corresponding changes. The normality of each variable was assessed by using the Kolmogorov–Smirnov test. Quantitative data was expressed by the mean, standard deviation, and difference between the means of two groups, which were tested by the Mann–Whitney U test, while for comparing more than two groups the Kruskal–Wallis H test was used. Univariate and multivariate logistic regression analysis was carried out to identify factors affecting symptom worsening, and an adjusted Odds Ratio was calculated along with 95% CI. A ‘*p*’ value less than 0.05 was considered statistically significant. Age, sex, race, preprocedural KCCQ score, Left Ventricular End Systolic Volume (LVESV), Left Ventricular End Diastolic Volume (LVEDV), Effective Regurgitant Orifice Area (EROA), Pulmonary Artery Systolic Pressure (PASP), history of atrial fibrillation, and adherence to medications/revascularization strategies in the follow-up period (ACE/ARB, beta blockers/cardiac resynchronization therapy) were used in the multivariate regression analysis. LVEDV, LVESV, EROA, and PASP are echocardiographic parameters with a proven association with outcomes after MitraClip^®^ placement.

Ethical approval for the study: [2216293-1] OUTCOMES FOR MITRAL CLIP PLACEMENT: SINGLE-CENTER RETROSPECTIVE REAL-WORLD ANALYSIS: PRISMA HEALTH IRB: Exemption Category # 4.

## 3. Results

In the one-year follow-up period, 94 patients had symptom improvement, 45 patients were symptom-stable, and 85 patients experienced symptom worsening/death. The baseline patient characteristics are depicted in Table 1.

Pre-procedural echocardiographic parameters between the groups were compared, and variance was analyzed across the groups. The results are summarized in Table 2.

Analyzing the baseline echocardiographic parameters of diastolic dysfunction, significant variance was observed for the E/A ratio, MV deceleration time, and MV gradient. The E/A ratio in the symptom-worsening group was found to be significantly higher than symptom improvement/stable group, while the MV deceleration time was significantly lower in the symptom-worsening group.

The results of the univariate regression analysis between the pre-procedural echocardiographic parameters and association with symptom worsening are summarized in Table 3. In the univariate regression analysis, the E/A ratio and MV deceleration time had a significant association with symptom worsening.

Multivariate regression analysis was carried out by comparing the E/A ratio in the pre-procedural echocardiogram with the symptom-worsening group. The results are summarized in Table 4. The E/A ratio was found to have a significant association with symptom worsening in the multivariate regression analysis.

Area under the curve (AUC) was carried out between the pre-procedural E/A ratio and symptom-worsening group and is depicted in Figure 1.

Based on Liu’s method, the optimal cutoff of the E/A ratio to predict a worsening status was 2.09, which provided a sensitivity of 58% and specificity of 63%. The area under the curve at this threshold was 0.60, indicating modest discriminative ability.

## 4. Discussion

The mitral valve apparatus is a dynamic structure with complex interactions surrounding its anatomy. Pathophysiologic mechanisms leading to FMR can be broadly divided into ischemic and non-ischemic causes [1]. Irrespective of the etiology, the primary driver is the lateral displacement of the annulus causing defective tethering forces. The decreased vertical tension created across the leaflets during systole can also be affected due to contractile dysfunction [11]. This more often leads to Carpentier IIIb functional regurgitation (Figure 2). Left atrial dilatation due to long-standing atrial fibrillation could also lead to a pathology like FMR, presenting more often as a Carpentier type I functional regurgitation [11]. This explains how different pathophysiologic mechanisms can have an end common result of functional regurgitation across the mitral valve. Analyzing the studies and trials carried out on MitraClip^®^, meaningful patient outcomes occur when the regurgitation is the primary symptom driver as well as when the clip aids in reverse remodeling.

The role of diastolic function in the progression of secondary MR is not well studied. In subjects with an ejection fraction of less than 50%, the E/A ratio is often used as a marker of left ventricular diastolic filling (of LV relaxation and compliance). In MR, the elevated LA and LV pressures can lead to elevated pressure gradients, the overestimation of the MV peak E velocity and decreased LA function, and the underestimation of the MV A-velocity [12]. Hence, the E/A ratios are not used in the estimation of diastolic function in the presence of MR. While carrying out the echocardiographic estimation of the diastolic function can be difficult in the presence of MR, the role played by impaired LA/LV relaxation and compliance can have significant effects on end diastolic pressures [13]. Irrespective of whether the etiology (ischemic versus non ischemic) of the FMR or the MR itself predominates, abnormal diastolic function can worsen the end diastolic pressures. The extent of diastolic dysfunction that could prohibit the remodeling effects of mTEER has not been well documented in previous trials. The 2020 AHA/ACC guidelines recommend a LVESD (left ventricular end systolic dimension) < 7 cm for mTEER in SMR, a threshold for remodeling beyond which the clip is unlikely to provide meaningful outcomes [2]. However, diastolic function parameters, which can play a significant role in determining end diastolic pressures, have not been traditionally used in patient selection criteria.

The MV deceleration time and E/A ratio documented in the pre-procedural echocardiogram were the factors that had statistically significant variance across the symptom groups. The MV deceleration time is inversely related to mitral regurgitation severity in severe PMR. This happens due to the rapid equilibration of pressures between the LA and the LV [14]. In a retrospective analysis of 234 patients that underwent undersized mitral annuloplasty ring, it was noted that an MV deceleration time of < 140 ms was predictive of worse outcomes [15]. In our analysis, the mean MV deceleration time in the symptom-worsening group was 176.88 ms (164.14–189.62) and had a mean difference of −28.44 ms (95% CI: −48.37–−8.51). A lower MV deceleration time was associated with symptom worsening in the univariate analysis with an Odds Ratio of 0.99 (0.89–0.99), implying an association between symptom worsening and a lower MV deceleration time. The association did not reach statistical significance in the multivariate regression analysis.

In our analysis, the pre-procedural E/A ratio had significant variance between the groups, with a *p*-value of 0.02 noted in the ANOVA. Running the chi-square test for the pre-procedural E/A ratio, the symptom-worsening group was found to have a significantly higher E/A ratio (mean difference 0.515, 95% CI: −0.88–−0.14, and *p*-value: 0.0067). In the multivariate regression analysis, a higher E/A ratio had an association with the symptom-worsening group, which reached statistical significance (aOR: 1.61 CI: 1.13–2.29). Age, sex, baseline KCCQ scores, LVEDV, LVESV, EROA, PASP, history of atrial fibrillation, medication adherence, and trans-mitral gradient were used in the regression analysis. Pre-procedural MV peak E velocity and MV deceleration time, despite having significant variance across outcome groups, were avoided in the regression analysis due to collinearity with the E/A ratio. In the ROC analysis, the higher E/A ratio had an AUC (area under the curve) of 0.603. The optimal cut off E/A ratio to predict symptom worsening was 2.094, which proved a sensitivity of 58% and specificity of 63%. In a retrospective analysis involving 102 patients, a higher peak E velocity was found to have significant association with Regurgitant Fraction in FMR [16,17]. In this perspective, it can be argued that a higher E/A ratio could be reflective of the regurgitation severity. Although an arbitrary cut-off value for the E/A ratio may not be used for patient selection due to subnormal sensitivity, E/A ratio would still serve as a marker for LV relaxation and compliance, irrespective of the severity of regurgitation.

Left atrial volume index (LAVi) is yet another marker for the LV diastolic function and can be a critical marker of LA remodeling in MR. Elevated LAVi > 40 mL/m^2^ can be an independent predictive factor of worse outcomes in primary MR. Lavi also correlates with the severity of MR. As MR severity increases, so does LAVi, reflecting the chronic volume overload and subsequent LA dilation [18]. This relationship is evident in both chronic and acute MR settings, where LA distensibility and function are significantly impacted [19,20]. In clinical practice, LAVi is used to guide the management of primary MR. The American College of Cardiology/American Heart Association (ACC/AHA) guidelines recommend considering mitral valve surgery in asymptomatic patients with severe primary MR and LAVi > 60 mL/m^2^, even if left ventricular function is preserved, due to the high risk of adverse outcomes [21]. In our analysis, no significant difference was observed in the pre-procedural LAVi across the symptom groups. The trans-mitral gradient before mTEER implantation is another well studied parameter as a predictor for clip success. Studies have shown that appropriate mTEER placement with reduced EROA and a trans-mitral gradient < 5 mmHg was a predictor of better outcomes [22]. Boerlage-van Dijk et al. found that the mean mitral valve gradient increased from 3.0 ± 1.6 mm Hg intraprocedural to 4.3 ± 2.2 mm Hg post procedurally and further increased during exercise, indicating that operators should be cautious about the potential for elevated gradients post-procedure [23]. Additionally, Neuss et al. reported that a gradient exceeding 5 mm Hg post implantation was associated with poorer long-term outcomes, including higher all-cause mortality and heart failure hospitalization [24]. In our analysis, the symptom groups did not differ statistically in terms of the pre- or post-procedural mean mitral valve gradient or the increase in gradient after mTEER implantation.

STUDY LIMITATIONS: Although the initial patient pool had a sample size comparable to the large- scale trials in the past, after applying the exclusion criteria, the sample size decreased. A larger sample size would have improved the strength of the findings. Although the E/A ratio was found to have a significant association with the symptom-worsening group in the multivariate regression analysis, a cut-off value for the E/A ratio above which optimal patient outcomes could be predicted was not derived from the study. Another limitation is the fact that other echocardiographic parameters like the E/e’ ratio were not included in the analysis because of limitations in echocardiographic data retrieval. The follow-up period was one year, and a longer follow-up period could have potentially added more meaningful insights. Although a fraction of the patients had a follow-up period over 1 year, the study was limited to one-year follow up for unifying the patient sample.

## 5. Conclusions

The E/A ratio, MV deceleration time, and trans-mitral gradient obtained in the pre-procedural echocardiogram had significant variance across the outcome groups. While the MV deceleration time and E/A ratio had a significant association with symptom worsening in the univariate analysis, only the E/A ratio in the pre-procedural echocardiogram had a significant association in the multivariate regression analysis. The area under the curve (AUC) analysis between the E/A ratio and the symptom worsening ROC value was 0.6307.

## Figures and Tables

**Figure 1 medicina-61-01092-f001:**
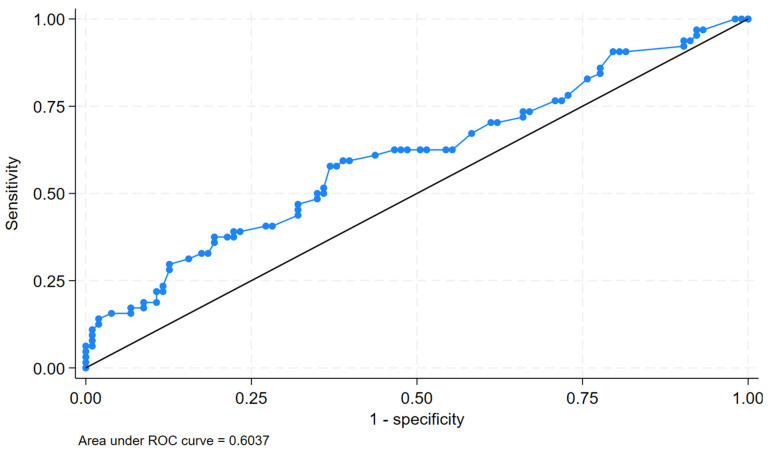
Area under the curve (AUC) analysis between pre-procedural E/A ratio and symptom worsening.

**Figure 2 medicina-61-01092-f002:**
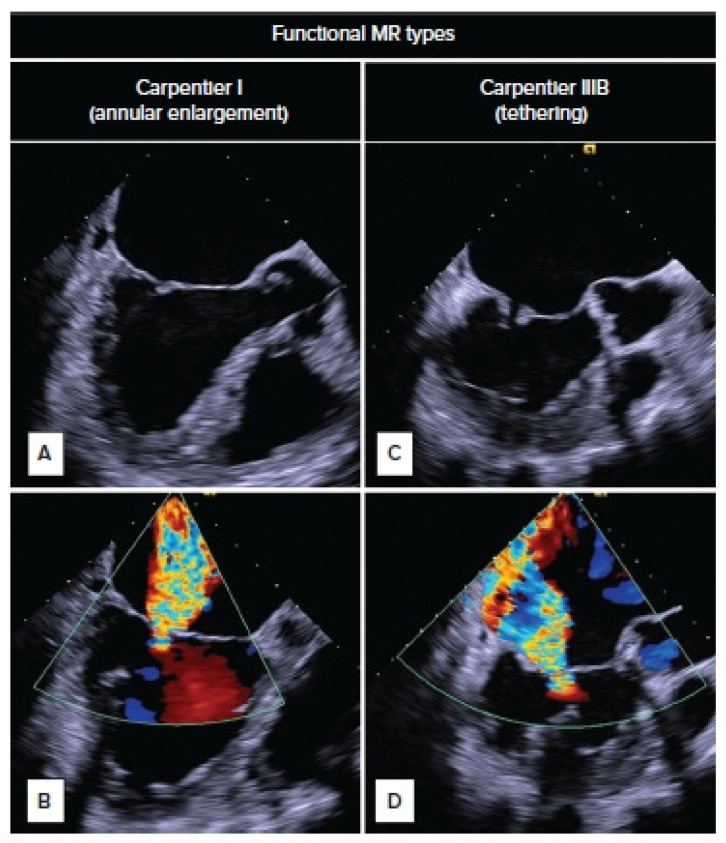
Carpentier classification: (**A**) Type1: Normal Leaflet Motion. (**B**,**C**) Excessive leaflet motion, such as prolapse or flail leaflets, often due to papillary muscle rupture, chordae rupture, or redundant chordae. Seen more often in atrial fibrillation and non-ischemic cardiomyopathy. (**D**) Type IIIb: Restricted motion primarily during systole, commonly associated with left ventricular dilation, papillary muscle dysfunction, or tethering of the leaflets. Seen in Ischemic cardiomyopathy.

**Table 1 medicina-61-01092-t001:** Baseline patient characteristics.

	Symptom-Improvement Group (94)	Symptom-Stable Group (45)	Symptom-Worsening/Death Group (85)
Mean Age (Years)	71.53 (68.98–74.07)	74.24 (70.42–78.06)	70.01 (67.15–72.86)
Sex	Male: 54 (57.44%)	Male: 21 (46.66%)	Male: 63 (74.11%)
	Female: 40 (42.55%)	Female: 24 (53.33%)	Female: 22 (25.88%)
Race	White: 63 (67.02%)	White: 34 (75.56%)	White: 39 (45.88%)
	Black: 31 (32.98%)	Black: 9 (20.00%)	Black: 43 (50.59%)
Participation in Cardiac Rehabilitation (%)	33 (37.50%)	9 (22.50%)	6 (7.50%)
History of Atrial Fibrillation (%)	62 (68.89%)	30 (69.77%)	60 (74.07%)
COPD Prevalence (%)	34 (37.78%)	22 (51.16%)	34 (43.04%)
CKD Prevalence (%)	52 (55.32%)	20 (44.44%)	57 (67.06%)
History of Myocardial Infarction (%)	36 (40.00%)	22 (53.66%)	23 (29.11%)
Adherence to ACE/ARB/Entresto (%)	64 (71.11%)	17 (40.48%)	50 (61.73%)
Adherence to Beta Blocker Therapy (%)	71 (78.89%)	41 (95.35%)	74 (92.50%)
Cardiac Resynchronization Therapy Post Procedure (%)	38 (42.22%)	16 (38.10%)	31 (38.75%)

**Table 2 medicina-61-01092-t002:** Pre-procedural echocardiographic parameters and variance across the groups.

Pre-Procedural ECHO Parameters	Group	Mean	SD	95% CI for Mean		*p* Value
				Lower Bound	Upper Bound	
Left Atrial Volume Index (mL/m^2^)	Symptom improvement	62.200	26.7069	54.765	69.635	0.15
	Symptom-stable	53.016	24.1662	44.715	61.318	
	Symptom-worsening/death	60.792	35.9249	51.512	70.073	
E/A Ratio	Symptom improvement	1.909	0.9194	1.681	2.137	0.02
	Symptom-stable	1.729	0.8088	1.460	1.999	
	Symptom-worsening/death	2.360	1.5389	1.975	2.744	
MV Deceleration Time\ (ms)	Symptom improvement	201.54	67.944	186.01	217.07	0.03
	Symptom-stable	212.50	80.833	186.65	238.35	
	Symptom-worsening/death	176.88	52.232	164.14	189.62	
MV Gradient (mm Hg)	Symptom improvement	1.90	1.012	1.69	2.11	<0.01
	Symptom-stable	2.77	2.125	2.11	3.42	
	Symptom-worsening/death	1.90	0.799	1.72	2.08	

MV: mitral valve, ECHO: echocardiogram, and E/A: peak E velocity/peak A velocity.

**Table 3 medicina-61-01092-t003:** Association of pre-procedural echocardiographic parameters with symptom-worsening group.

Pre-Procedural Echocardiographic Parameters	Odds Ratio of Association with Symptom-Worsening Group	95% Confidence Interval	*p*-Value
E/A Ratio	1.51	1.10–2.06	0.010
MV Deceleration time (ms)	0.99	0.98–0.99	0.008
MV Gradient (mm Hg)	0.822	0.63–1.07	0.148
Left Atrial Volume Index (mL/m^2^)	1.003	0.99–1.01	0.583

MV: mitral valve, ECHO: echocardiogram, and E/A: peak E velocity/peak A velocity.

**Table 4 medicina-61-01092-t004:** Multivariate regression analysis for association of E/A ratio with symptom worsening.

Pre-Procedural Echocardiographic Parameters	Odds Ratio of Association with Symptom-Worsening Group	95% Confidence Interval	*p*-Value
E/A ratio	1.61	1.13–2.29	0.008

E/A: peak E velocity/peak A velocity.

## Data Availability

The original contributions presented in this study are included in the article. Further inquiries can be directed to the corresponding author.

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
