# Peer review of "Diastology and MitraClip® Outcomes: Multicenter Real-World Evidence Study"

_medicina, 2025, doi:10.3390/medicina61061092_

Round 1
Reviewer 1 Report
Comments and Suggestions for Authors
Dear Authors,
This is an interesting study regarding the diastolic outcomes of MitraClip placement in patients with functional mitral regurgitation. The study investigates a knowledge gap in the MitraClip placement area. However, there are some concerns about it.
I believe the title could be rearranged to a more precise and informative one. For example, what do you affirm by stating “a real-world evidence study”?
Why is diastology emphasized regardless of the MitraClip outcome?
Title examples:
- MiraClip outcomes and diastology: a retrospective multicenter study
- MitraClip diastolic outcomes: a retrospective multicenter study
- Retrospective multicenter study evaluating MitraClip outcomes and diastology
The abstract’s English quality is inferior. I recommend it to be thoroughly refined. I would also suggest revising the rest of the manuscript’s English, as some minor edits are needed.
I prefer the introduction to be a bit shorter, focusing more on the knowledge gap, rationalizing the study objectives.
The ethical approval of your study should be mentioned in the materials and methods section.
I understand that you included patients treated from 2021 to 2024 for FMR with at least one-year follow-up; however, why did you not include earlier patients to present a more longitudinal follow-up?
The result section should mention the demographics of the included patients.
It would be better to demonstrate the proportion of patients included from each center.
I did not notice any table summarizing baseline characteristics (age, sex, comorbidities, and…) of the included patients, except for echocardiographic parameters.
The discussion is a bit unstructured.
The opening paragraph of the discussion should be a summary of key findings. Further, the interpretation of the results should be implied. Prior studies' results should be compared to the results and appropriately discussed, highlighting the path to the conclusions.
The study's strengths and limitations should also be mentioned. For example, why was a long-term follow-up not presented?
The conclusion should be completely rewritten. It's more like it’s presenting the more important results. The conclusion should consist of the take-home messages of the study.
Comments on the Quality of English Language
The quality of the English language should be properly revised with the help of an English language expert.
Author Response
Title has been changed
The abstract’s English quality is inferior. I recommend it to be thoroughly refined. I would also suggest revising the rest of the manuscript’s English, as some minor edits are needed. REPLY: abstract has been revied.
I prefer the introduction to be a bit shorter, focusing more on the knowledge gap, rationalizing the study objectives. Reply: the introduction part has been made more concise and shorter, focussing just on the aspects of the study
The ethical approval of your study should be mentioned in the materials and methods section. Reply: added
I understand that you included patients treated from 2021 to 2024 for FMR with at least one-year follow-up; however, why did you not include earlier patients to present a more longitudinal follow-up? REPLY: the patient data was only available starting from 2021 ( correlated with the hospital system switching to EPIC): could not get patient data prior to that
The result section should mention the demographics of the included patients. Reply: will add
It would be better to demonstrate the proportion of patients included from each center. Reply: majority of patients were from one center ( Richland), and the same operators performed the procedure at these centers ( names not included due to IRB regulations )
I did not notice any table summarizing baseline characteristics (age, sex, comorbidities, and…) of the included patients, except for echocardiographic parameters. Reply: will add along with the baseline patient characteristics
The discussion is a bit unstructured.
The opening paragraph of the discussion should be a summary of key findings. Further, the interpretation of the results should be implied. Prior studies' results should be compared to the results and appropriately discussed, highlighting the path to the conclusions. REPLY: discussion section has been changed. Kindly review
The study's strengths and limitations should also be mentioned. For example, why was a long-term follow-up not presented?
The conclusion should be completely rewritten. It's more like it’s presenting the more important results. The conclusion should consist of the take-home messages of the study. REPLY: strengths and limitations have been added. Conclusion section changed
The opening paragraph of the discussion should be a summary of key findings. Further, the interpretation of the results should be implied. Prior studies' results should be compared to the results and appropriately discussed, highlighting the path to the conclusions.
REPLY: the opening paragraph focussed on what we already know about diastology in MR, and the functional implications to add context to why diastology is being studied. In terms of the main factors compared, including E/A ratio and the MV deceleration time, implications of the results as well comparitive previous studies are presented
Reviewer 2 Report
Comments and Suggestions for Authors
This manuscript studied the association between pre procedural echocardiographic parameters of diastolic function and one-year outcomes after MC placement. Authors found pre-procedural E/A ratio had significant association with symptom worsening after multivariate regression analysis. It is well written and organized. However, authors might benefit from addressing the following points:
Major points:
- At line 197, is a sensitivity of 58% and specificity of 63% sufficient for prediction?
- The description of data shown in result section is missing. So are conclusions that should be drawn from data shown.
- Some texts in discussion section should be moved to result section. Discussion should not cite any of figures or tables.
Minor points:
- There are two figure1.
- What is panel C in figure1 at line 213?
Author Response
- At line 197, is a sensitivity of 58% and specificity of 63% sufficient for prediction? REPLY: is not a good predictive tool as a specific cut off value: will change the discussion part
- The description of data shown in result section is missing. So are conclusions that should be drawn from data shown.
- Some texts in discussion section should be moved to result section. Discussion should not cite any of figures or tables. REPLY: conclusions and discussions have been redone
- There are two figure1.
- What is panel C in figure1 at line 213? REPLY: changes made
- The description of data shown in result section is missing. So are conclusions that should be drawn from data shown. All the data shown in the results section is described in the discussion
- At line 197, is a sensitivity of 58% and specificity of 63% sufficient for prediction? A comment has been made in the conclusion
- What is panel C in figure1 at line 213? B and C are same physiology as per carpentier: changed
Round 2
Reviewer 1 Report
Comments and Suggestions for Authors
The quality of the English language still needs to be improved.
Please mention the publication year of "the current AHA/ACC guidelines..." wherever mentioned in the manuscript.
Although the results are summarized in tables provided within the manuscript, it is important to mention the key findings in the text.
In the discussion section, where you have stated "However, diastolic function parameters, that can play a significant role in determining end diastolic pressures, have not been traditionally used in patient selection criteria," should be clarified. Are you referring to your own experiences, or is it based on others' diagnostic methods?
The conclusion's modifications are not satisfactory. The structure is not acceptable. It includes unnecessary information that distracts the reader from the key messages.
Comments on the Quality of English Language
The quality of the English language still needs to be improved.
Author Response
The quality of the English language still needs to be improved.
Reply: language editing has been done. Please review
Please mention the publication year of "the current AHA/ACC guidelines..." wherever mentioned in the manuscript.
Reply: year 2020 has been added
Although the results are summarized in tables provided within the manuscript, it is important to mention the key findings in the text.
reply: findings of the results section have been added after the tables in the results section
In the discussion section, where you have stated "However, diastolic function parameters, that can play a significant role in determining end diastolic pressures, have not been traditionally used in patient selection criteria," should be clarified. Are you referring to your own experiences, or is it based on others' diagnostic methods?
Reply: this is based on the fact that we could not find any large scale studies assessing diastology in mitraclip outcomes based on our chart review ( scholar, scopus, pubmed and embase)
The conclusion's modifications are not satisfactory. The structure is not acceptable. It includes unnecessary information that distracts the reader from the key messages.
reply: conclusions section have been edited. Kindly review
Reviewer 2 Report
Comments and Suggestions for Authors
Thanks for authors' reply. Also, my concerns are shown below.
- Some of my points raised before were not addressed.
- The description of data should be in the result section not the discussion section.
Author Response
The description of data should be in the result section not the discussion section.Reply: description of results have been added in paragraphs below the tables in the results section